Water absorption through salivary gland type I acini in the blacklegged tick, Ixodes scapularis

Kim Donghun
Maldonado-Ruiz Paulina
Zurek Ludek
Park Yoonseong ypark@ksu.edu
Department of Entomology, Kansas State University , Manhattan , KS , United States of America
Oppert Brenda
Electronic publication date: 2017 Oct 31
Publication date: 2017
Volume: 5
Electronic Location ID: e3984
Received 2017 Jun 29; Accepted 2017 Oct 12
Copyright: ©2017 Kim et al.
Copyright year: 2017
Copyright holder: Kim et al.
License: This is an open access article distributed under the terms of the Creative Commons Attribution License, which permits unrestricted use, distribution, reproduction and adaptation in any medium and for any purpose provided that it is properly attributed. For attribution, the original author(s), title, publication source (PeerJ) and either DOI or URL of the article must be cited.
License URL: https://creativecommons.org/licenses/by/4.0/

Keywords: Salivary gland, Vector, Rhodamine, Water balance

Funding: National Institute of Health R01AI090062 This work was supported by the National Institute of Health; Grant Number (R01AI090062). The funders had no role in study design, data collection and analysis, decision to publish, or preparation of the manuscript.

==============================
Tick salivary glands play critical roles in maintaining water balance for survival, as they eliminate excess water and ions during blood feeding on hosts. In the long duration of fasting in the off-host period, ticks secrete hygroscopic saliva into the mouth cavity to uptake atmospheric water vapor. Type I acini of tick salivary glands are speculated to be involved in secretion of hygroscopic saliva based on ultrastructure studies. However, we recently proposed that type I acini play a role in resorption of water/ions from the primary saliva produced by other salivary acini (i.e., types II and III) during the tick blood feeding phase. In this study, we tested the function of type I acini in unfed female Ixodes scapularis. The route of ingested water was tracked after forced feeding of water with fluorescent dye rhodamine123. We found that type-I acini of the salivary glands, but not type II and III, are responsible for water uptake. In addition, the ingestion of water through the midgut was also observed. Injection or feeding of ouabain, a Na/K-ATPase inhibitor, suppressed water absorption in type I acini. When I. scapularis was offered a droplet of water, ticks rarely imbibed water directly (5%), while some approached the water droplet to use the high humidity formed in the vicinity of the droplet (23%). We conclude that during both on- and off-host stages, type I acini in salivary glands of female Ixodes scapularis absorb water and ions.

Introduction

Maintaining water balance in terrestrial arthropods is crucial for survival. Water uptake occurs via various routes, including the anus, cuticle, air vapor, and direct drinking (Dunbar & Winston, 1975; Edney, 1977; Kahl & Knülle, 1988; McMullen, Sauer & Burton, 1976; Rudolph & Knulle, 1974). Ixodid ticks can capture atmospheric water molecules using hygroscopic saliva in the microenvironment where the relative humidity is higher than critical equilibrium activity (McMullen, Sauer & Burton, 1976; Rudolph & Knulle, 1974). Different types of salivary gland acini have been suggested to have different roles in the water balance in Ixodid ticks (Coons et al., 1994; Kim, Šimo & Park, 2014; Krolak, Ownby & Sauer, 1982; Megaw & Beadle, 1979; Needham, Rosell & Greenwald, 1990).

Ixodid ticks are the major vectors that transmit various pathogens causing diseases including babesiosis, anaplasmosis, and Lyme disease. In the United States, Lyme disease is the best well-known tick-borne disease transmitted by the blacklegged tick, Ixodes scapularis. In addition to their importance in human health, ticks can be an interesting model system for studying homeostasis of water. This is because of their unique ecological properties; their long blood feeding (up to two weeks) comparing to that of other blood feeding arthropods, such as mosquitoes; and their long duration of fasting (several months) when off the host. Tick salivary glands play critical roles in water balance. In female ticks, a pair of salivary glands located in the anterolateral regions of the tick body, consists of three different types of acini: types I/II/III. Type I acini are located on the anterior portion of the main salivary duct, type II acini are on the secondary branches, and type III acini are located on the most distal end tertiary branches of the salivary glands. During blood feeding, salivary glands play critical roles in eliminating excess water and ions obtained from a large amount of blood (Kaufman & Phillips, 1973; Tatchell, 1967). Autocrine/paracrine dopamine signaling orchestrates an influx of solute into type-II/-III acini via dopamine receptor (D1) and efflux of solute through the salivary duct by pumping/gating the acini via invertebrate D1-like dopamine receptor (InvD1L) (Kim, Šimo & Park, 2014; Šimo et al., 2014; Šimo et al., 2011). Additionally, neuropeptides (SIFamide and MIP) innervating the basal region of salivary gland acini type-II/-III are thought to be involved in controlling salivary secretion (Šimo et al., 2014; Šimo, Koči & Park, 2013; Šimo, Žitňan & Park, 2009).

During fasting, ticks are known to secrete hygroscopic saliva into the mouth cavity to uptake atmospheric water vapor and maintain the water balance (Gaede & Knülle, 1997; McMullen, Sauer & Burton, 1976; Rudolph & Knulle, 1974). In the dry environment with lower relative humidity (RH) than critical equilibrium activity (CEA), secreted crystalline hygroscopic saliva is then crystalized, which leads to absorption of water molecules at the RH higher than CEA, and ticks then imbibe the fluid (McMullen, Sauer & Burton, 1976; Rudolph & Knulle, 1974). Type I acini have been thought to be important in this process because the size of type I acini remains constant in both, off- and on-host, stages; while type-II/-III acini are enlarged in the on-host stages. In addition, in an ultrastructural study, lamellate cells, cells consisting of type I acini, have extensive interdigitating plasma membranes facing hemolymph and large numbers of mitochondria between the infolds, which are features similar to the nasal salt glands of marine birds (Doyle, 1960) that produce hygroscopic solution for withdrawing atmospheric water molecules. Therefore, type I acini have been generally thought to be the site of secretion for hygroscopic saliva (McMullen, Sauer & Burton, 1976; Rudolph & Knulle, 1974).

However, in this study, we found that the type I acini are indeed the site of water and ion absorption in the off-host phase, as determined by the absorption function of type I acini via Na/K-ATPase in the off-host phase of female I. scapularis. In investigations of fasting ticks, we found that water uptake occurs through both type I acini and the midgut, which is partially dependent on ouabain-sensitive Na/K-ATPase.

Materials & Methods

Ticks and dehydration of ticks

Unfed adult blacklegged ticks (I. scapularis) were obtained from the tick rearing center at Oklahoma State University (Stillwater, OK, USA). Ticks were kept in an incubator at 28 °C and 98% relative humidity (RH) until the experiments. Partially dehydrated unfed female ticks were prepared by placing the ticks in the dehydration condition (28 °C and RH 25%) for 36 h.

Feeding rhodamine123 (Rh123) and imaging fluorescence

To investigate the route of water intake in ticks, 1.28 µL of 1 mM rhodamine123 in water (Sigma-Aldrich, St. Louis, MO, USA) was filled in a microcapillary tube (Diameter ∼0.11 mm, Sigma-Aldrich, St. Louis, MO, USA) and the tube was placed onto the mouthparts of dehydrated unfed female ticks. Ticks were allowed to ingest Rh123 under rehydration conditions (RH 98%) at room temperature for 30 min. After ingestion, we quantified the ingested amount and traced the locations of fluorescence signal in tick organs, specifically in the salivary glands and the gut diverticular. The volume ingested was calculated by the equation for cylinder volume (V = πr2h). For calculation of volume, the reduced length (h) of fluid in microcapillary tubes was measured under a grid microscope and analyzed with r2 = 0.0127. The locations of fluorescence in the internal organs were identified after dorsal integument of ticks was removed by a surgical scalpel. Images were captured using a camera (DFC400) attached to a stereo microscope (M205FA; Leica, Heerbrugg, Switzerland). Salivary glands were subsequently dissected out, fixed in 4% paraformaldehyde at room temperature for 30 min, washed in PBST (0.1% Triton X-100), and imaged on a confocal microscope (LSM700; Zeiss, Oberkochen, Germany).

Injection/pre-ingestion of ouabain or Hank’s saline buffer

To investigate the physiological function of Na/K-ATPase, we injected or orally introduced ouabain as a Na/K-ATPase blocker and Hank’s saline buffer as control. We injected 50 nL of 100 µM ouabain (Sigma Aldrich, St. Louis, MO, USA) or Hank’s saline into dehydrated (36 hr) unfed female ticks with a Nanoliter 2010 injector controlled by Micro4 (WPI, Sarasota, FL, USA). Ticks injected by either ouabain or Hank’s saline were placed in dehydration conditions for an additional 30 min; then, these ticks were subsequently fed fluid containing 1 mM Rh123 in a microcapillary tube under the rehydration conditions in a water saturated glass jar (RH 98%) for 30 min.

For pre-ingestion experiments, dehydrated unfed female ticks were offered ouabain (100 µM) or Hank’s saline in a microcapillary tube for 30 min. After the 30-min pre-ingestion of ouabain (varied between 0.1 and 0.15 µL), the micropipettes were replaced by micropipettes filled with Rh123 under the rehydration condition for an additional 30 min. After injection and pre-ingestion experiments, the ingested volume and fluorescence signal were quantified and imaged as described in a previous section.

Natural water drinking of dehydrated unfed female ticks

To examine whether questing I. scapularis drink water, dehydrated and unfed female ticks were exposed to 5 µL of water in the center of a 50 mm x 9 mm airtight petri dish (Falcon, NY, USA) at room temperature for 1 h. We counted the number of ticks that approached and ingested the natural water through their mouthparts.

Statistical analyses

The significant difference of each experiment was found with either a Student’s t-test (p = 0.05) or chi-square test. The t-test was used to analyze statistical differences between control and experimental groups of Rh123 ingestion assays (n = 25 total), pre-ingestion assays (n = 11 in each treatment), and injection assays (n = 5 in each treatment). The Chi-square test was used to analyze the frequency of Rh123 fluorescence of type I acini from pre-ingestion assays with Hank’s saline buffer and ouabain.

Results and Discussion

Water absorption through type I acini of salivary glands

Studies of the ultrastructure of type I acini have suggested that type I acini play a critical role in the secretion of hygroscopic saliva to capture atmospheric water vapor (Binnington, 1978; Kahl & Alidousti, 1997; McMullen, Sauer & Burton, 1976; Megaw & Beadle, 1979; Needham, Rosell & Greenwald, 1990). In this process, the type I acini are generally thought to be the direct site for producing hygroscopic saliva (Krolak, Ownby & Sauer, 1982; Megaw & Beadle, 1979; Needham, Rosell & Greenwald, 1990), while the function of the salivary glands and midgut for direct water absorption is speculated as the site to uptake water (McMullen, Sauer & Burton, 1976). However, a study focused on ultrastructural changes of type I acini in different humidities contradicts the function of type I acini in direct production of hygroscopic saliva. Dehydration of ticks results in an orthodox configuration (inactive form) of mitochondria, while rehydration changed the mitochondria to a condensed configuration (active form in rehydration) in the type I lamellate cells (Needham, Rosell & Greenwald, 1990; Needham & Coons, 1984). This is in contradiction to the prediction that type I acini participate in active production of hygroscopic saliva. In this study, we provide clear evidence that type I acini function in direct water absorption.

To visualize the routes of water absorption in fasting ticks, we performed forced feeding of fluorescent tracer Rh123 for 30 min using a microcapillary tube on dehydrated unfed female ticks. There were large individual variations in the ingested amount of solution in the range of 0.052–0.448 µL. The majority of the tested individuals (64%) were positive for fluorescence in both cell bodies of type I acini (but not in type II/III acini) and the gut diverticular (Figs. 1A and 2), while other individuals (36%) with low amount of ingestion (<0.15 µL) lacked fluorescence in type I acini (Fig. 1B), but presented only in the gut diverticula (Figs. 1B and 2).

Figure 1 Fluorescence in gut diverticular and salivary glands after feeding water with the tracer dye rhodamine 123 (Rh123).

(A) Example of fluorescence positive in both salivary glands and gut diverticular. (B) Example of fluorescence positive only in gut diverticular. Empty arrow heads indicate gut diverticular. Solid arrow head indicates salivary glands. Asterisks indicate auto-fluorescence of hindgut and rectal sac, which were confirmed in previous observation (Kim et al., 2016). Scale bar equals 0.5 mm.

Figure 2 Consumed volume of water containing rhodamine 123 (Rh123) after a forced feeding for 30 min.

The consumed volume is compared for the ticks with the fluorescence and without fluorescence in the type I acini. Each symbol indicates ingested amount of Rh123 of individual dehydrated female I. scapularis tick after a forced feeding for 30 min (n = 25). Boxes indicate range data from 25% to 75%. The horizontal line in the box is for median and the diamond mark is mean. Whiskers with lines indicate 99% and 1% of data. The significant difference (p = 0.05, asterisk) was found in Student T-test.

This result was similar to our previous observation (Kim et al., 2016) with some important differences. The previous study showed all tested ticks were positive for fluorescence in the type I acini, while some lacked fluorescence in the midgut, which is in contrast to the results in this study with partially positive type I acini. We speculated that the difference was caused by two reasons, the degree of dehydration of the ticks (12 h vs. 36 h in this study) and the longer feeding duration (immediately after the drinking water (<15 min vs. 30 min drinking in this study). Long dehydration times likely resulted in inactivation of the type I acini in some individual ticks, considering it was demonstrated with the inactive form of mitochondria in type I acini found in 24 h dehydrated ticks (Needham, Rosell & Greenwald, 1990). Longer pre-conditioning of dehydration and a longer duration of forced-drinking in this study compared to previous studies likely allowed for drinking through the gut, while some individuals lacked water uptake through the salivary glands.

In individuals that ingested high levels of rhodamine123, fluorescence was not only found in the type I acini and gut but it also permeated to the hemolymph. This observation suggests that the ticks actively uptake water through type I acini. In addition, the presence of tracer in the cytoplasm of type I acini cells and the hemolymph implies that the absorption mechanism likely includes active transport for the tracer dye Rh123, which is often used for experiments testing the roles of the membrane transporter (Forster et al., 2012; Jancis et al., 1993).

Na/K-ATPase in water absorption

We tested the role of Na/K-ATPase in water absorption using ouabain, which is a well-known Na/K-ATPase inhibitor. Previous studies described abundant Na/K-ATPase immunoreactivity in the type I acini (Kim et al., 2016; Needham, Rosell & Greenwald, 1990). Initially, we injected ouabain in the hemocoel where it likely inhibits Na/K-ATPases in the whole body. The injection was followed by forced feeding of Rh123 through a microcapillary tube to quantify the ingested volume and observe the fluorescence of salivary glands.

Ouabain injection significantly reduced the volume of water ingestion (Fig. 3). The ingested volume of ouabain-injected ticks was 28% of Hank’s saline-injected ticks (n = 5 for each). Most of the type I acini lacked tracer, which was only observed from the main duct in ouabain injections (Fig. 4B), while Hank’s saline injections showed Rh123 fluorescence in the type I acini (Fig. 4A).

Figure 3 Reduction in the volume of consumed water after ouabain injection.

Total ingested amount of water containing Rh123 for 30 min were compared between ticks injected either by Hank’s saline (n = 5) or ouabain (n = 5). Each symbol indicates ingested volume of individual dehydrated female I. scapularis tick. Boxes indicate data range from 25% to 75%. The horizontal line in the box is for median and the diamond mark is mean. Whiskers with lines indicate 99% and 1% of data. The significant difference (p = 0.05, asterisk) was found in Student T-test.

Figure 4 Tick salivary glands showing fluorescence in the type I acini after a forced feeding of water containing rhodamine 123 (Rh123).

Green and blue colors indicate rhodamine 123 and nuclei, respectively. (A) Salivary gland from Hank’s saline-injected ticks displayed green fluorescence in the type-I acini. (B) Salivary gland from Ouabain-injected ticks lacked green fluorescence in the type-I acini. Scale bars indicate 100 µm.

To achieve more specific inhibition of Na/K-ATPase in the type I acini of salivary glands, we treated ticks with ouabain by pre-ingestion. Therefore, pre-ingestion of ouabain for 30 min (uptaken volume varied between 0.1 and 0.15 µL) affected Na/K-ATPase in water uptake by type I acini and the midgut specifically. Ticks with ouabain pre-ingestion consumed 67% of the levels of control, Hank’s saline pre-ingested ticks. The frequency of individuals with Rh123 fluorescence in the type I acini was significantly lower in the ticks with pre-ingestion of ouabain than those with pre-ingestion of Hank’s saline (Fig. 5B; p = 0.004). Although the effect of pre-ingestion of ouabain on type I-mediated water uptake was less pronounced than the effect shown in the injection experiment, this was presumably due to the low dose reaching the target tissue. The broad pharmacological effects of ouabain may also include consequences of mitochondrial calcium deficiency or impairment of mitochondrial energy metabolism, which are also known activities of ouabain (Liu, Brown & O’Rourke, 0000; Roevens & De Chaffoy de Courcelles, 1990), in addition to the inhibitory effects on Na/K-ATPase in type I acini by the ouabain pre-ingestion. This result supports the significant roles of Na/K-ATPase in the type I acini-mediated water uptake.

Figure 5 Effects of ouabain pre-ingestion on the absorption function of type I acini.

(A) Amount ingested by the ticks that pre-ingested Hank’s saline (n = 11) and ouabain (n = 11). Each symbol indicates ingested volume of rhodamine 123 of individual dehydrated female I. scapularis. Boxes indicate data range from 25% to 75%. The horizontal line in the box is for median and the diamond mark is mean. Whiskers with lines indicate 99% and 1% of data. The significant difference (p = 0.05, asterisk) was found in Student T-test. (B) Comparison fluorescence observed from type-I acini between pre-ingestion with Hank’s saline and ouabain. The data were analyzed by Chi-Squared test (p = 0.004).

Based on the experiments with Rh123 and ouabain, we found ingested water flowing into both, the gut diverticular space and salivary glands. Dehydration levels of ticks influenced the route of water absorption via gut diverticular spaces and salivary glands or gut diverticular spaces only. Under severe dehydration, Na/K-ATPase mediated water uptake in type I acini might not be functional due to the inactive form of mitochondria (Needham, Rosell & Greenwald, 1990). Rehydration by water uptake via the gut diverticula likely activates Na/K-ATPase-mediated water uptake in type I acini.

Natural behaviors in water uptake

Finally, we investigated whether I. scapularis voluntarily drink water. We observed several different patterns of behavior when a water droplet was offered. The majority of dehydrated ticks were not attracted to water and randomly moved or stayed away from the water drop (72%, 31/43, Fig. 6A). However, some dehydrated ticks actively approached and stayed close to the water drop without directly contacting the water with their mouthparts (23%, 10/43, Figs. 6C and 6D). These ticks displayed two patterns: (i) they spread the front pair of legs toward the water drop and the extended legs and often directly touched the water drop (Fig. 6C); and (ii) they folded their first two pairs of legs and stayed close to the water drop (Fig. 6D). A small group of dehydrated ticks actively approached the water drop, placed the chelicera on top of the water surface, and drank the water (5%, 2/43, Fig. 6B and Video S1). In the third case, pulsatile water flow was observed between hypostome and chelicerates (Video S1).

Figure 6 Three different patterns of behavior observed from the dehydrated unfed female I. scapularis when a water drop is offered.

(A) Percent of different tick behavioral patterns; No attraction, Attracted & Stayed, and Drink water in pie chart. (B) Example of ticks drinking water via mouthpart. (C and D) Two sub-patterns of attracted & stayed, respectively. (C) Spreading the front pair of legs toward water drop and directly touching water drop. (D) Folding first two pairs of legs and stayed close to water drop.

Previous studies described individuals from the Ixodes genus that approached a water drop to use the high humidity formed on the vicinity of the drop, but they did not directly drink the water (Kahl & Alidousti, 1997; Lees, 1946; Yoder & Spielman, 1992). We found that, unlike the earlier study of I. ricinus and I. scapularis (Kahl & Knülle, 1988; Lees, 1946; Yoder & Spielman, 1992), under our experimental conditions, I. scapularis directly drank water in rare occasions (5%).

A model for the roles of type I acini in water balance

A previous study indicated that on-host ticks actively excreted an excess of salt and water through salivary glands by producing iso/hypo osmotic saliva (Kim et al., 2016). Sodium-rich primary saliva was produced by a dopamine-mediated electrochemical gradient in type II and III acini. In that study and the present study, immunohistochemistry revealed Na/K-ATPase in type II/III in addition to type I acini, indicating Na/K-ATPase was the major source for the electrochemical gradient in the formation of primary saliva (Fig. 7A). Based on the results of this study in the fasting tick, we propose that hygroscopic hyperosmolar saliva is formed as a result of shutting down the absorptive function of type I acini under severely dry conditions (Fig. 7B). Therefore, the hyperosmolar primary saliva formed from type II/III acini in the previous study was directly secreted without the Na/K-ATPase-mediated resorptive function of type I acini (Kim et al., 2016). Once water molecules are captured in the hygroscopic saliva, type I acini are the site for absorbing water from the diluted saliva (Fig. 7C). The ions (mainly Na+) used for generation of electrochemical gradient for water uptake in type I acini are likely recycled for secretory activity in the type II/III acini.

Figure 7 A model proposed for the function of type I acini in water absorption.

(A) During blood feeding, ticks secrete iso/hyposmotic saliva. Type II and III acini produce hyperosmotic primary saliva, and type-I acini subsequently reabsorb ions immediately before secretion. (B) During fasting in vegetation, ticks secrete hyperosmotic saliva to uptake water vapor from subsaturated air. Hyperosmotic saliva is mainly produced by type II and III acini. Nonfunctional type I acini is shown by gray X mark. (C) During fasting in dehydrated condition, captured water from air vapor is absorbed via type I acini, while type II and III acini have no function in water absorption. The acini figure is modified from Binnington (1978).

Cellular mechanisms of the tick salivary glands appear to be strikingly similar to that of insects (i.e., the American cockroach) (Hille & Walz, 2008). Na/K-ATPase in the apical surface of acinar peripheral cells in the insect is similar to that in the apical surface of types II and III acini in ticks (Kim et al., 2016). The resorptive function of type I acini with Na/K-ATPase in the basolateral infolding is the same as the duct cells in insect salivary glands with a similar subcellular location of Na/K-ATPase. This configuration facilitates the production of primary saliva in the distal part of salivary glands (types II and III in tick and peripheral cells in the American cockroach), which is followed by resorption of ion/water in the proximal part of the salivary glands (type I and duct).

A previous study successfully showed the phenotype for RNAi of Na/K-ATPase that resulted in failure in the full engorgement and reduced egg numbers in oviposition (Karim et al., 2008). Based on the RNAi of Na/K-ATPase in our laboratory followed by accessing the degree of suppression using real time PCR and immunohistochemical staining of Na/K-ATPase, partial suppression of Na/K-ATPase from synganglion and salivary glands was associated with phenotype failure in full engorgement. However, the immunohistochemistry of Na/K-ATPase suggested that knock down of Na/K-ATPase immunoreactivities in type I acini did not occur, while partial knock down of the immunoreactivities in types II/III acini were observed (Figs. S1–S2). Our RNAi results implied that knocking down a gene product could be also tissue/target specific, and suggested that type I acini Na/K-ATPase was difficult to knock down by RNAi due to the long half-life of the protein.

The absorptive function of type I acini found in this study reconcile with results of previous studies. Our study is important in: (a) identifying a route for drug delivery that may be useful for physiological studies, (b) uncovering a novel mechanism of water absorption in ticks and that may be also common in other arachnid species, and (c) application of this new knowledge for tick management.

Supplemental Information

Video S1 Direct water drinking observed in a dehydrated unfed female I. scapularis tick

Video record showing a tick drinking water through the mouth cavity between hypostome and chelicerates. Chelicerates cover half of hypostome but not all. Pulsatile movement of water flow is shown in the cavity filled with water by capillary force.

Click here for additional data file.

Figure S1 Na/K-ATPase immunoreactivities were observed in type I/II/III of Hank’s saline buffer injected unfed female salivary glands. Arrow heads indicated immunoreactivities of Na/K-ATPase

Red indicated positive immunoreactivities of Na/K-ATPase. Overview image of salivary glands (A). Close image focusing on type II and III acini (B & C). Scale bar indicated 50 um.

Click here for additional data file.

Figure S2 Na/K-ATPase immunoreactivities were observed in only type I of dsRNA-Na/K-ATPase injected unfed female salivary glands

Arrow heads indicated immunoreactivities of Na/K-ATPase. Red indicated positive immunoreactivities of Na/K-ATPase. Overview image of salivary glands (A). Close image focusing on type II and III acini (B & C). Scale bar indicated 50 um.

Click here for additional data file.

Supplemental Information 1 Raw data for Figs. 2, 3, 5 and 6

Click here for additional data file.

Supplemental Information 2 Raw confocal picture for Fig. S1A

Click here for additional data file.

Supplemental Information 3 Raw confocal picture for Fig. S1B

Click here for additional data file.

Supplemental Information 4 Raw confocal picture for Fig, S1C as a Tif format

Click here for additional data file.

Supplemental Information 5 Raw confocal picture for Fig. S2A

Click here for additional data file.

Supplemental Information 6 Raw confocal picture for Fig. S2B

Click here for additional data file.

Supplemental Information 7 Raw confocal picture for Fig. S2C

Click here for additional data file.

This study is contribution no. 18-176-J from the Kansas Agricultural Experiment Station.

Additional Information and Declarations

Competing Interests

Author Contributions

Data Availability

The authors declare there are no competing interests.

Donghun Kim conceived and designed the experiments, performed the experiments, analyzed the data, wrote the paper, prepared figures and/or tables, reviewed drafts of the paper.

Paulina Maldonado-Ruiz conceived and designed the experiments, performed the experiments, reviewed drafts of the paper.

Ludek Zurek conceived and designed the experiments, contributed reagents/materials/analysis tools, wrote the paper, reviewed drafts of the paper.

Yoonseong Park conceived and designed the experiments, analyzed the data, contributed reagents/materials/analysis tools, wrote the paper, prepared figures and/or tables, reviewed drafts of the paper.

The following information was supplied regarding data availability:

The raw data is included as Supplemental Files.

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
