# Peer review of "Water absorption through salivary gland type I acini in the blacklegged tick, Ixodes scapularis"

_PeerJ, doi:10.7717/peerj.3984_

## Round 0.1 · original submission · Major Revisions

· Academic Editor

Major Revisions

As some of the reviewers have noted, some editing will help to improve the message of the manuscript. For example, here are suggestions for editing of the abstract:
"Tick salivary glands play critical roles in maintaining water balance for survival, as they eliminate excess water and ions during blood feeding on hosts. In the long duration of fasting during the off-host period, ticks secrete hygroscopic saliva into the mouth cavity to uptake atmospheric water vapor. Type I acini of tick salivary glands are speculated to be involved in secretion of hygroscopic saliva based on ultrastructure studies. However, we recently proposed that type I acini play a role in resorption of water/ions from the primary saliva produced by other salivary acini (i.e., types II and III) during the tick blood feeding phase. In this study, we tested the function of type I acini in unfed female Ixodes scapularis. The route of ingested water was tracked after forced feeding of water with the fluorescent dye rhodamine123. We found that type-I acini of the salivary glands, but not type II and III, are responsible for water uptake. In addition, the ingestion of water through the midgut also was observed. Injection or feeding of ouabain, a Na/K-ATPase inhibitor, suppressed water absorption in type I acini. When I. scapularis was offered a droplet of water, ticks rarely drank imbibed water directly (5%), while some approached the water droplet to use the high humidity formed in the vicinity of the droplet (23%). We conclude that during both on- and off-host stages, type I acini in salivary glands of female I. scapularis absorb water and ions."

Please consider comments from reviewers, and edit the manuscript accordingly. Thank you for submitting the article to PeerJ.

Regards,

Brenda

Reviewer 1 ·

Basic reporting

Please see comments to author. Minor adjustments are needed.

Experimental design

No Comments

Validity of the findings

Please see comments to author. Minor adjustments are needed.

Additional comments

Reviewer comments to the submitted manuscript entitled, “Water absorption through salivary gland type I acini 1 in the blacklegged tick, Ixodes scapularis” by D Kim and colleagues.
In the present study, the authors perform a number of molecular and physiological analyses to demonstrate the mechanisms of water absorption and rehydration through the tick salivary gland. The authors go on to suggest that type 1 acini are responsible for water balance, which is a nice addition to Y. Park’s previous work focused on the neuroendocrinology of tick salivation. The authors suggest the Na+-K+-ATPase protein may represent a target site for tick control, but this reviewer thinks the potential for exploiting this target as a site of control is limited. This reviewer does not see any glaring errors in the research design or the presentation of the data.

The manuscript is well written with very few grammatical errors, repetitions, or incorrectly cited references. The data are clearly presented. There are a few minor revisions that are suggested:

1. Previous work by D Kim and Y Park has shown evidence that the Na+-K+-ATPase pump is involved in ion secretion rates. It is likely that the water secretion is due to the development of an osmotic gradient that is derived from an electrochemical gradient that facilitates the transport of anions. However, the authors show the Na+-K+-ATPase is expressed in the type II and III cells for ion secretion whereas this study suggests the Na+-K+-ATPase is responsible for water balance in type 1 cells. This reviewer suggests discussing the expression patterns of Na+-K+-ATPase and the potential implications for osmotic regulation and maintenance.
2. Line 85: This reviewer suggests including the final concentration of rhodamine in the feeding studies versus the volume and starting concentration.
3. Line 117: was this water labeled with Rho123?
4. Methods: It would be beneficial to include a statistics section that highlights the number of replicates, statistical analyses used, etc
5. The data presented in lines 174-184 assume that oaubain is inhibiting a membrane expressed Na+-K+-ATPase pump. However, this molecule is not completely specific for Na+-K+-ATPase and has been shown to alter calcium cycling, mitochondria functioning, etc. Either of these will most certainly alter the function of the salivary gland acini and should be addressed with additional studies or, at a minimum, a more thorough discussion.
6. Lines 231-240: Historically, RNAi has had difficulties penetrating through the basement membrane of tick salivary glands and has been shown to be marginally effective at reducing mRNA constructs. Therefore, the assumption that the inability to reduce Na+-K+-ATPase mRNA due to a long half-life is a large assumption and should be softened. Also, it would be of great benefit to include a figure showing the RNAi knockdown in the synganglion lading to reduced feeding.

Reviewer 2 ·

Basic reporting

The manuscript by Kim et al., "Water absorption through salivary gland type I acini in the blacklegged tick, Ixodes scapularis" is well written and addresses an important topic, water balance in ticks. This work builds on these authors previous studies. However there are concerns to this current version.

The general concern is that its unclear in this manuscript why this was conducted. Authors should clearly state the objectives of this study, how data in the manuscript accomplished these objectives. What is the implication or contribution of these data to tick physiology?

Some specific objectives include
1. Explain the formula that was used to calculate the volume of water that was imbibed by ticks.
2. Clearly state controls
3. Results section lines 195-199 did not make sense, what does natural behaviors in drinking water?

Experimental design

Experiments not well described, please explain your controls

Validity of the findings

Difficult to determine in absence of well explained controls

Additional comments

Explain objectives and how data here met those objectives

Reviewer 3 ·

Basic reporting

no comment

Experimental design

This research is properly designed and described

Validity of the findings

Conclusions are well stated otherwise I have no comments

Additional comments

An important contribution to the literature on salivary glands of ixodid ticks. The only error I could find is the incorrect authorship for a cited paper which should read as follows:
Needham G, Rosell R. Greenwald L. and Coons, LB

---

## Round 0.2 · Minor Revisions

· Academic Editor

Minor Revisions

Dear Yoonseong,

I have edited your manuscript, mostly for clarity and verb tense, but there is one section (A model for the roles of type I acini in water balance) which is an important summary and there are a few sentences that were not clear. Please incorporate the edits as appropriate. In addition, the acknowledgment section needs to be edited. I cannot upload a word file to edit, so I append a PDF and PeerJ staff will send you a word file with track changes.

---

## Round 0.3 · accepted · Accept

· Academic Editor

Accept

Thank you for submitting to PeerJ!